# First Report of *Enterocytozoon hepatopenaei* Infection in Pacific Whiteleg Shrimp (*Litopenaeus vannamei*) Cultured in Korea

**DOI:** 10.3390/ani11113150

**Published:** 2021-11-04

**Authors:** Bo-Seong Kim, Gwang-Il Jang, Su-Mi Kim, Young-Sook Kim, Yu-Gyeong Jeon, Yun-Kyeong Oh, Jee-Youn Hwang, Mun-Gyeong Kwon

**Affiliations:** Aquatic Disease Control Division, National Fishery Products Quality Management Service, 216 Gijanghaean-ro, Gijang-eup, Gijang-gun, Busan 46083, Korea; gijang2@korea.kr (G.-I.J.); sumikim@korea.kr (S.-M.K.); eprose@hanmail.net (Y.-S.K.); jsjs56928@gmail.com (Y.-G.J.); oskal@korea.kr (Y.-K.O.); jinihwang@korea.kr (J.-Y.H.); mgkwon@korea.kr (M.-G.K.)

**Keywords:** EHP, hepatopancreatic microsporidiosis, growth disorder, parasite, ssu rRNA gene, microspora

## Abstract

**Simple Summary:**

We here report the first detection of *Enterocytozoon hepatopenaei* (EHP) in cultured Pacific whiteleg shrimp (*Litopenaeus vannamei*) with a growth disorder in Korea. The histopathology, electron microscopy and DNA phylogeny indicated that the EHP is a unique strain first recorded in Korea. This finding highlights the need for closer monitoring and surveillance to control EHP in aquaculture to prevent disastrous economic losses.

**Abstract:**

The consumption of cultured crustaceans has been steadily increasing, and Pacific whiteleg shrimp (*Litopenaeus vannamei*) are major cultivated invertebrates worldwide. However, shrimp productivity faces a variety of challenges, mainly related to outbreaks of lethal or growth retardation-related diseases. In particular, hepatopancreatic microsporidiosis caused by the microsporidian parasite *Enterocytozoon hepatopenaei* (EHP) is an important disease associated with growth retardation in shrimp. Here, we report the detection of EHP through histopathological, molecular and electron microscopy methods in the hepatopancreas of Pacific whiteleg shrimp with growth disorder in a South Korean farm. Phylogenetic analysis showed a clade distinct from the previously reported EHP strains isolated in Thailand, India, China and Vietnam. An EHP infection was not associated with inflammatory responses such as hemocyte infiltration. Although EHP infection has been reported worldwide, this is the first report in the shrimp aquaculture in Korea. Therefore, an EHP infection should be managed and monitored regularly for effective disease control and prevention.

## 1. Introduction

The Pacific whiteleg shrimp (*Litopenaeus vannamei*) is the most widely farmed species for human consumption [1]. As a major aquaculture crustacean, the farm production scale of Pacific whiteleg shrimp was 5,446,216 tons and USD 32,191 million globally in 2019 [2], and was 8124 tons and USD 125 million in Korea in 2020 [3]; however, this production is insufficient to meet the market demand in the country [4]. Stable production is essential to meet consumption needs. A major contributor to production losses is disease. Various diseases have been reported in Asia that cause mortality or slow growth disorder during shrimp aquaculture [5], including hepatopancreatic microsporidiosis (HPM), which is a slow growth disorder caused by an infection of the microsporidian parasite *Enterocytozoon hepatopenaei* (EHP). HPM was first reported as a growth disorder symptom in the Black tiger shrimp (*Penaeus monodon*) in Thailand in 2004 [6], and the causative pathogen was identified as a novel parasite based on histopathological, electron microscopic and phylogenetic analyses in 2009 [7]. To date, Black tiger shrimp, Pacific whiteleg shrimp, and Blue shrimp (*Litopenaeus stylirostris*) have been reported as the main hosts infected with EHP [5,6,8], and major outbreaks have been reported in Vietnam, India, Brunei, China, Indonesia, Malaysia, Venezuela and Australian [8,9,10,11,12].

EHP is a parasite with two life stages: an extracellular stage with an effective (mature) spore phase in the lumen of the digestive tubule, and numerous intracellular spore-forming stages that colonize the digestive epithelial cells of the shrimp hepatopancreas [13]. Although a severe EHP infection does not cause high mortality in shrimp, it is considered to be a wasting disease that contributes to economic loss due to a decrease in production at shrimp farms [5].

Although there have been reports of EHP detection in shrimp imported to Korea, there has been no report on pathogen isolation or associated disease occurrence in domestic shrimp farms [14]. In April 2020, a significant growth delay of Pacific whiteleg shrimp was noted compared to previous years at a farm located in Ganghwa-gun, Incheon, Korea. When shrimps were cultured from post-larva for 12 weeks, the normal shrimp were double in size compared to the shrimp with growth disorders. In this study, we performed phylogenetic, histopathological and electron microscopic analyses of these samples, which consistently identified an EHP infection. To our knowledge, this is the first report of the EHP strain infecting Pacific whiteleg shrimp in Korea. This work can, therefore, offer guidance for the prevention, control and management of HPM outbreaks.

## 2. Materials and Methods

### 2.1. Sampling

In April 2020, a significant growth delay occurred in a Pacific whiteleg shrimp farm in Ganghwa-gun, Incheon, Korea. Observation of the field conditions did not reveal any white feces in the culture seawater, which is a common clinical sign of severe infection with EHP [9]; however, empty intestines were found among individuals with growth retardation. Therefore, shrimps (total 20 individuals) ranging from 3.7 to 9.3 cm were randomly collected from the farm and transported alive to the laboratory in a 20-L seawater tank for molecular biological, histological and electron microscopic analyses to identify the causal pathogen.

### 2.2. Polymerase Chain Reaction (PCR) Detection and Prevalence of EHP

The hepatopancreas extracted from the 20 shrimp were pooled into four samples of five shrimp each for PCR analysis. DNA was extracted from sampled shrimp using a 5-min DNA/RNA extraction kit (BioFactories, Monrovia, CA, USA), according to the manufacturer’s instructions. A nested PCR amplification of the small subunit ribosomal RNA (ssu rRNA) gene of the EHP was performed in two rounds of nested PCR, according to a previous study [15]. For the nested PCR of the spore wall protein (SWP) gene, SWP_1F and SWP_1R primers were used for the first PCR reaction, and SWP_2F and SWP_2R primers were used for the second PCR reaction [16]. The amplified DNA was visualized using the QIAxcel Advanced Capillary Electrophoresis System (Qiagen, Hilden, Germany). Direct sequencing of the purified PCR products for the EHP gene was performed using EHP-specific primers (ENF779 and ENR779) in an applied Biosystems sequencer at SolGent (Seoul, Korea). The ssu rRNA gene sequence of strain NFQS-EHP1 was obtained and compared against the GenBank databases using Blastn [17]. The ssu rRNA gene sequence of NFQS-EHP1 obtained from the shrimp was deposited in the GenBank database (accession number MZ819965). Phylogenetic analysis was performed using the MEGA X program [18]. A phylogenetic tree based on the neighbor-joining method [19] was constructed using bootstrap analysis of 1000 replications.

To measure the prevalence of EHP infection at the shrimp farm, the nested PCR results were input into the prevalence package in R software with the test sensitivity set to 95–100% and the test specificity set to 90–95% [13,20].

### 2.3. Histopathology and Transmission Electron Microscopy (TEM)

Histopathological and ultrastructural microscopic analyses were performed on the extracted hepatopancreas samples from Pacific whiteleg shrimp exhibiting delayed growth. For light microscopic observation, the hepatopancreas was fixed in a 10% neutral buffered formalin, dehydrated, embedded in paraffin wax, sectioned at 4 μm and stained with hematoxylin and eosin (H & E) and Giemsa solution. For ultramicroscopic observation, small pieces (1 mm^3^) of the hepatopancreas were fixed in 2.5% glutaraldehyde, post-fixed in 1% osmium tetroxide, embedded with an epoxy-embedding kit (Sigma-Aldrich, St. Louis, MO, USA), dehydrated through a graded acetone series, sectioned at 70–90 nm and stained with uranyl acetate and lead citrate. The sections were examined using a Libra120 transmission electron microscope.

## 3. Results

### 3.1. Macroscopic Observation of Shrimp with Delayed Growth

The phenotypes of the shrimp samples were very different between normal individuals and those with growth disorder. The normal and abnormal shrimps were 8.5–9.3 cm and 3.7–5.6 cm in length, respectively (Figure 1A). The intestines of the shrimp with delayed growth were empty in contrast to those of the shrimp with normal growth (Figure 1B).

### 3.2. Nested PCR and Prevalence of EHP Infection

Three of the four pooled samples tested positive for EHP in nested PCR reactions. As shown in Figure 2, the phylogenetic tree of ssu rRNA genes demonstrated that the EHP sequence obtained from the shrimp with delayed growth (designated NFQS-EHP1) clearly separated from previously reported EHP sequences and formed its own monophyletic clade. Using Blast analysis, the ssu rRNA gene similarity between NFQS-EHP1 and other EHP sequences in the phylogenetic tree were 99.7 to 99.9%. The SWP gene similarity between this study and others in the top ten blast matches were 100% in the GenBank database by using BLAST analysis (Appendix A). The prevalence based on the test conditions (sensitivity and specificity settings) and the EHP detection rate was calculated to be 25.5% (95% confidence interval: 3.3–56.0%).

### 3.3. Microscopic Observation of Shrimp with Delayed Growth

Histopathologically, a large amount of EHP was observed in the digestive tubules of the hepatopancreas. Eosinophilic EHP showed a size of 1.4–1.7 μm in H & E staining and appeared as a 5.0–10.0-micrometer cluster of aggregated EHP (Figure 3A,B). This distribution was more evident with Giemsa staining. EHP was abundantly distributed inside the epithelial cells of the digestive tubule, and was also detected outside of the cells, indicating that the parasite had leaked out due to necrosis and rupture (Figure 3C,D). Most of the individuals with growth disorders showed a serous exudate in the hepatopancreatic interstitial tissue, but only a small number of hemocytes were found adjacent to the digestive tubule.

### 3.4. TEM Observation

The TEM of the hepatopancreas revealed mature EHP ranging from 1.4 to 1.7 μm in length with typical characteristics of coiled polar filaments, posterior vacuoles, polaroplasts, endospores (53.0 nm), electron-dense exospores (12.0 nm) and nuclei (Figure 4).

## 4. Discussion

Histopathological and molecular biological analyses of Pacific whiteleg shrimp with growth disorders revealed the presence of an EHP infection in a shrimp farm in Ganghwa-gun, Korea. Phylogenetic analysis indicated that the obtained isolate from this farm can be considered a new strain, distinct from previously reported EHP in other countries, which is classified as an independent clade. This is the first case of EHP infection detected in Pacific whiteleg shrimp in Korea.

Numerous EHPs, ranging from 1.4 to 1.7 μm in size, were observed both inside and outside the epithelial cells of the digestive tubule with H & E and Giemsa staining. In contrast to serious inflammation signs, such as granuloma formation and hemocyte infiltration, that commonly accompany the bacterial and fungal infections of shrimp [21,22,23,24], only a mild serous exudate was observed in the hepatopancreatic interstitial tissue of EHP-infected shrimp, without evidence of massive hemocyte infiltration. A heavy infection of EHP in the hepatopancreas, which is the center of the nutrient metabolism in shrimp [25], can cause not only physical failure of nutrient absorption but also hepatopancreatic necrosis, enabling the release of mature EHP. High levels of aspartate transaminase (AST) and alanine transaminase (ALT) in the hemolymph were related to necrosis and extensive damage in hepatopancreatic cells due to a severe infection of EHP, and were observed in naturally and experimentally EHP-infected shrimp [26].

EHP is considered a wasting disease that continuously causes disturbances in nutrient absorption and storage without inflammatory changes. Recently, a proteomic study identified biomarkers linked to a growth hormone disorder in shrimp with growth retardation [27]; changes in these growth hormones can result in decreased immunity and increased susceptibility to other diseases [28,29]. In addition, a complex infection with EHP and bacteria can induce granuloma and septic hepatopancreatic necrosis, as well as mortality [30]. This suggests that co-infection can increase mortality due to secondary infections. Furthermore, an EHP infection has been reported to increase the susceptibility to acute hepatopancreatic necrosis disease, which has a high mortality rate, therefore posing a major risk [31].

Changes in environmental factors such as ammonia and nitrites can promote the development of HPM. The safe concentrations of total ammonia nitrogen and nitrite for shrimp culture are known to be less than 0.834 and 0.328 mg/L, respectively [32]. Factors that increase EHP infection in shrimp farms include ammonia and nitrite levels above 1 mg/L [33]; feeding on prey organisms infected with EHP, such as *Artemia salina* [34] and polychaetes [35]; and horizontal transmission through the breeding seawater [36]. We estimated the prevalence of EHP at the shrimp farm to be approximately 25.5% (95% confidence interval: 3.3–56.0%). If the disease is already rampant, it is difficult to remove EHP because of the life-cycle within the epithelial cells of the hepatopancreas and there is currently no available drug for treatment [34]. Thus, it is essential to manage the disease appropriately to effectively control the infection and prevent it spreading. To prevent the occurrence of HPM in shrimp farms, efforts should be made to focus on the root causes with proper management of the environment, inactivation of EHP in prey through frozen storage [12], regular non-lethal methods of EHP monitoring of the brood stock [37] and routine disinfection using safe disinfectants such as chlorine [12].

## Figures and Tables

**Figure 1 animals-11-03150-f001:**
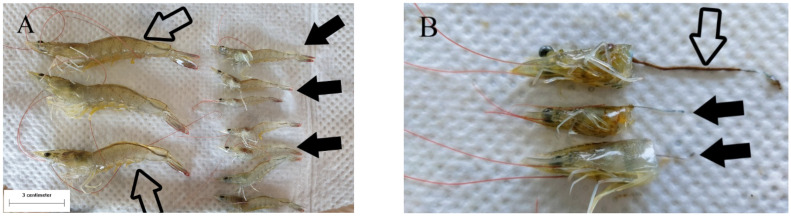
Gross observations of shrimp with different growth patterns. (**A**) Pacific whiteleg shrimp with normal growth (open arrows) and with delayed growth (solid black arrows). (**B**) Empty intestines (solid black arrows) in contrast to the intestine filled with digestive substances (open arrow).

**Figure 2 animals-11-03150-f002:**
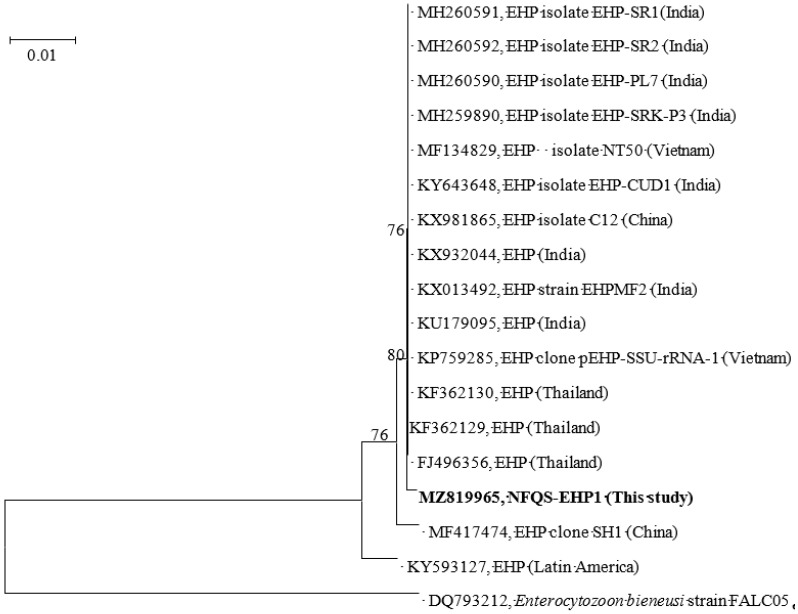
The phylogenetic tree was constructed using the neighbor-joining method, using MEGA software (version X; http://www.megasoftware.net (accessed on 18 August 2021)). All reference sequences were acquired from the GenBank database (http://www.ncbi.nlm.nih.gov/genbank (accessed on 18 August 2021)). *Enterocytozoon bieneusi* (DQ793212) was used as the outgroup. Only bootstrap values above 70% are shown (1000 resamplings) at branch points. Bar, 0.01 substitutions per site.

**Figure 3 animals-11-03150-f003:**
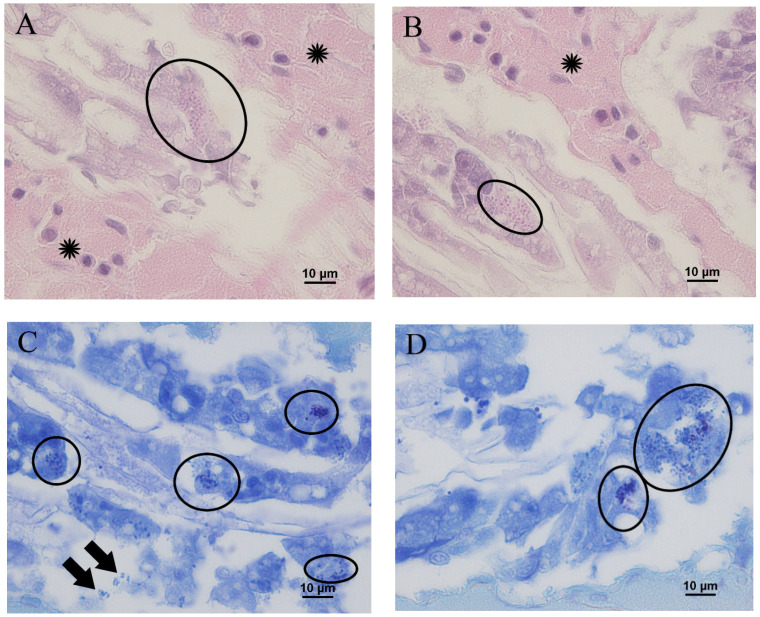
Hepatopancreas of Pacific whiteleg shrimp infected with EHP. (**A**,**B**) Hematoxylin and eosin staining revealing numerous eosinophilic EHP in the cytoplasm of epithelial cells in the hepatopancreas (ellipses) and edema in the interstitial tissue (asterisk). (**C**,**D**) Giemsa staining revealing numerous EHP (purple) in the cytoplasm of epithelial cells in the hepatopancreas (ellipses), along with EHP released into the lumen of the digestive tubule (arrows).

**Figure 4 animals-11-03150-f004:**
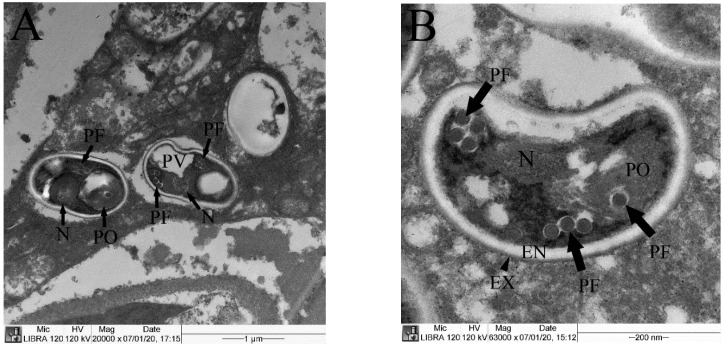
Transmission electron microscopy images of the hepatopancreas from Pacific whiteleg shrimp infected with EHP. (**A**) Mature spores showing a polar filament (PF), polaroplast (PO), nucleus (N) and posterior vacuole (PV). (**B**) Mature spore showing an endospore (EN), exospore (EX), polar filament (PF), polaroplast (PO) and nucleus (N).

## Data Availability

All data are contained within the article. Please contact the corresponding author for additional data requests.

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
