# Peer review of "First Report of Enterocytozoon hepatopenaei Infection in Pacific Whiteleg Shrimp (Litopenaeus vannamei) Cultured in Korea"

_animals, 2021, doi:10.3390/ani11113150_

Round 1
Reviewer 1 Report
Dear Editor,
This article reported the first detection of the microsporidian parasite Enterocytozoon hepatopenaei (EHP) in cultured Pacific whiteleg shrimp (Litopenaeus vannamei) with growth disorder in Korea. A distinguishable EHP parasite was histopathologically identified using Giemsa staining, and was characterized by electron microscopy. Phylogenetic analysis showed that the EHP sequence of the isolate clearly separated from previously reported EHP sequences, and formed its own clade. This finding highlights the need for closer monitoring and surveillance to control EHP in aquaculture for preventing disastrous economic losses in Korea. In my opinion, there were some additional experiments needed, this paper is not suitable for publishing in Animals so far.
Major comments:
1) This article suggested the EHP isolated from Korea was a new isolate, but only a phylogenetic tree for SSU was shown. It is necessary to supplement the results of Blastn between this SSU gene with others previous reported. I found the identity was up to 99.87% for the SSU gene from this article with MH260591, MH260590, and so on which was in Fig. 2. So, it is hard to confirm the EHP isolated in this article was a novel isolate. Therefore, in order to prove this result, it is suggested to add some other analysis results. For example, the sequence alignment results of other conserved genes, such as SWPs, PTPs or other conserved genes in EHP.
2) Line 81, which gene was designed for ENF779 and ENR779 primer? Please provide the Genbank number for this gene.
Minor comments:
I would suggest adding a bar in Fig. 1A.
Reviewer 2 Report
Comments
Manuscript animals-1366114 is a good study investigating parasite Enterocytozoon hepatopenaei infection in Pacific whiteleg shrimp in Korea. Ms is well written which is easy to follow; however, some major issues are found that need to be addressed before ms is suitable for publication. Issues and comments are listed as follows:
Abstract
26 - It would be better just to mention “India” since West Bengal is one of states in India.
Introduction
36 - should be “(Litopenaeus vannamei)” Make it consistent as what the authors write in the simple summary and abstract!
- is or are?? See lines 18-19!
38 - should be common name instead of its scientific name!
46 - see the previous comment in line 36!
48-49 - see the previous comment in line 38!
60-61 - see the previous comment in line 38!
61 - “Pacific whiteleg shrimp” could be omitted!
60-62 - Could the authors provide data showing how significant growth delay of shrimp in the farm such as culture parameters (it could be growth rate, biomass, weight, or length)?
- references for this statement should be included!
65 - see the previous comment in line 38!
M&M
73 - the condition of shrimp (normal and abnormal) should be specified!! See the previous comment regarding growth retardation in lines 60-62!
77-78 — The way of sample collected should be explained. Were they collected from dead or live shrimp? How did the authors treat the animal during collection of sample? Where were the samples kept?
78 - “to detect EHP” can be omitted!
80-84 - state the fragment size of each pair of primers used!
92-94 - I would recommend including the similarity analysis of sequence obtained in this study compared to other sequences from the data bases (by using BLAST analysis for example).
95-97 - the authors need to explain a bit detail about how to test the prevalence of infection (sensitivity and specificity). How did the author perform these tests considering that the samples were pooled in this study?
Results
111-112 - mention clearly the different phenotypes of shrimp from both groups (normal and abnormal groups). See the previous comment in lines 60-62, and 73!
115 - “(L. vannamei)” can be omitted!
119 - it seems that samples from normal growth group were also positive EHP. State this clearly!
1120-124 - this results can be supported by similarity analysis of sequences! See the previous comment in lines 92-94!
144 - “(L. vannamei)” can be omitted!
153 - “(L. vannamei)” can be omitted!
Discussion
157 - should be “Pacific whiteleg shrimp”
171-172 - this sentence is not really connected with what author discuss in the previous statement.
184-185 - Here, the authors discuss about the environmental factors affecting HPM disease in the farm. It would be nice if the author can provide data showing the water quality parameters (ammonia and nitrite) in the shrimp farms in Ganghwa-gun where the samples taken.
Conclusion
- the conclusion needs to be rewritten since it is not really accurate especially regarding to EHP infection in shrimp with growth retardation. Please see the previous comments related to this.
198-199 - “(L. vannamei)” can be omitted!
199 - should be “Pacific whittle shrimp”
201 - this statement is not necessary, and can be omitted!
202 - should be “Pacific whiteleg shrimp”
Reviewer 3 Report
Line 104: last word "fixed" should be "post-fixed"
Line 165 and 168 "hemocytes infiltration" is better English if expressed as "hemocyte infiltration".
Line 190 "because of the inner life cycle in epithelial cells" should be "because of the life-cycle within epithelial cells".
Reference 15, pages are not "1-10" but "139" as in
Tangprasittipap, A., Srisala, J., Chouwdee, S. et al. The microsporidian Enterocytozoon hepatopenaei is not the cause of white feces syndrome in whiteleg shrimp Penaeus (Litopenaeus) vannamei. BMC Vet Res 9, 139 (2013). https://doi.org/10.1186/1746-6148-9-139
Figure 3 A,B is barely acceptable. It is very pale probably because the Eosin solution was exhausted when stained.
Round 2
Reviewer 1 Report
For this manuscript (ID: animals-1366114.v1), I reviewed it and raised three questions. But two of them did not get an appropriate reply from the author in the revised version. So, I suggested rejecting this article and resubmitting it after modification.
Point 1: It was proved that the EHP isolated in the manuscript was a new isolate.
From the SWP gene blast result, it was shown that SWP sequence of EHP isolated in the manuscript was the same as others, suggesting it was not a new isolate. The author said “deleted the words related with novel in the revised manuscript (lines 23-24,121-124)”, but there was “this strain is thus considered to be novel” in Line 27.
Point 3: Adding a bar in Fig. 1A.
Although the author said they have added the bar in Fig. 1A, there was no bar in Fig. 1A.
Reviewer 2 Report
Comments-animals-1366114-v2
Manuscript animals 1366114 has been improved by the authors according to the previous comments. However, not all issues are well responded which need more information and confirmation. The authors should address the rest of issues as listed below to meet the ms is fit for publication.
56-8 - state duration period of rearing for shrimp to reach those sizes!
70 - should be “…in 20L seawater tank..”
84-6 why did the authors add other new primers here?? For what? State this!!
123-6 - as the previous comments in lines 84-6, why did the authors use two genes of EHP for similarity analysis?
- mention other species (some of them) which has highest similarity with the species in this study!
171-3 - then, the authors should modify the sentence by including the information stated in the response letter!
183-4 - Regarding the environmental conditions (ammonia and nitrite), the authors could use secondary data from other reports if they are available.
195 - Previous comment about conclusion is not completely addressed by the authors. I would recommend rewriting the conclusion by including information related to EHP infection in shrimp with growth retardation based on the this study.
